# Temporal Effects of Quercetin on Tight Junction Barrier Properties and Claudin Expression and Localization in MDCK II Cells

**DOI:** 10.3390/ijms20194889

**Published:** 2019-10-02

**Authors:** Enrique Gamero-Estevez, Sero Andonian, Bertrand Jean-Claude, Indra Gupta, Aimee K. Ryan

**Affiliations:** 1Department of Human Genetics, McGill University, Montreal, QC H4A 3J1, Canada; egameroestevez@gmail.com (E.G.-E.); guptalab@gmail.com (I.G.); 2Research Institute of the McGill University Health Centre, Glen Site, Montreal, QC H4A 3J1, Canada; bertrandj.jean-claude@mcgill.ca; 3Division of Urology, McGill University, Montreal, QC H4A 3J1, Canada; sero.andonian@mcgill.ca; 4Department of Medicine, McGill University, Montreal, QC H4A 3J1, Canada; 5Departments of Pediatrics, McGill University, Montreal, QC H4A 3J1, Canada

**Keywords:** tight junctions, claudins, kidney stones, paracellular transport, ion reabsorption, quercetin

## Abstract

Kidney stones affect 10% of the population. Yet, there is relatively little known about how they form or how to prevent and treat them. The claudin family of tight junction proteins has been linked to the formation of kidney stones. The flavonoid quercetin has been shown to prevent kidney stone formation and to modify claudin expression in different models. Here we investigate the effect of quercetin on claudin expression and localization in MDCK II cells, a cation-selective cell line, derived from the proximal tubule. For this study, we focused our analyses on claudin family members that confer different tight junction properties: barrier-sealing (Cldn1, -3, and -7), cation-selective (Cldn2) or anion-selective (Cldn4). Our data revealed that quercetin’s effects on the expression and localization of different claudins over time corresponded with changes in transepithelial resistance, which was measured continuously throughout the treatment. In addition, these effects appear to be independent of PI3K/AKT signaling, one of the pathways that is known to act downstream of quercetin. In conclusion, our data suggest that quercetin’s effects on claudins result in a tighter epithelial barrier, which may reduce the reabsorption of sodium, calcium and water, thereby preventing the formation of a kidney stone.

## 1. Introduction

Kidney stones affect 10% of the population and recur in 50% of adults. They are associated with renal failure in children and adults. They cause extreme pain and have a significant financial burden to society [1,2,3]. Given these morbidities, surprisingly little is known about why stones recur and how they can be prevented. Most stones are calcium-based with calcium oxalate more frequently observed than calcium phosphate stones [1]. Kidney stones result from the accumulation of salts along the kidney nephron, where an important factor in salt reabsorption is the epithelial tight junction barrier. Tight junctions are the most apical junction between apposing cells, where they compartmentalize the apical and basolateral intercellular space and regulate the passive paracellular movement of water and solutes [4]. The ion specificity of the tight junction barrier is determined by the claudin family of tetraspanin proteins, which contains close to 30 members [5,6,7]. Claudins can bind hetero- and homotypically with claudins in the same cell through their transmembrane domains or to claudins in the apposing cell through their extracellular loops [6]. The combination of claudins expressed within an epithelium determines the tightness and selectivity of the tight junction barriers [8].

The claudin composition of tight junctions varies along the different segments of the nephron and, consequently, ions and salts are differentially reabsorbed in the different nephron segments [8]. Reabsorption of calcium from the urine filtrate predominantly occurs in two segments: the proximal tubule reabsorbs 70% and the thick ascending limb reabsorbs 25% of the calcium. Several claudin family members are implicated in calcium reabsorption. A common sequence variant in CLDN14 and rare mutations in CLDN16 and -19, all of which are expressed in the thick ascending limb, are risk factors for the formation of calcium-based kidney stones in humans [9,10,11]. Both Cldn2 and -10 null mice develop nephrocalcinosis [12,13], while Cldn16 and -19 knockdown mice develop hypercalciuria [14,15]. Cldn7 is also essential for salt reabsorption: Cldn7 null mice die shortly after birth due to salt loss and dehydration [16]. Although not known to participate in ion exchange, Cldn3 has been shown to form a complex with Cldn16 and -19 in the thick ascending limb, which is essential for calcium and magnesium reabsorption in this segment [17]. Cldn4 has a critical role in chloride reabsorption in the kidney collecting duct where it forms a pore with Cldn8 [18]. The role of Cldn1 in ion reabsorption is less clear; however, it is present in different kidney segments and may be important in diabetic nephropathy [19]. Because claudins are essential for sodium, calcium, and water reabsorption in the nephron, targeting this claudin-based epithelial barrier may be a successful approach to decrease salt reabsorption and prevent kidney stone formation.

The flavonoid quercetin prevents kidney stone formation in a rat model [20]. Quercetin alters claudin expression in the Caco-2 intestine-derived cell line and in LLC-PK1 cells, an anion-selective cell line derived from the renal proximal tubule [21,22,23,24]. To date, no one has studied the effect of quercetin in a cation-selective cell line that models an epithelial barrier which is permeable to calcium and sodium. We hypothesize that quercetin may prevent kidney stones through its effects on claudins at tight junctions in the epithelial barrier of the nephron. For our study, we used Madin-Darby Canine Kidney cells (MDCK II), which are cation-selective and derived from the proximal tubule where the majority of sodium and calcium reabsorption occurs. MDCK II cells express Cldn1, -2, -3, -4, and -7 [25,26], which allows us to investigate the effect of quercetin on claudins with different properties: barrier-sealing (Cldn1, -3 and -7), cation-selective (Cldn2), or anion-selective (Cldn4) within a cation-selective cell line [27,28,29,30]. The MDCK II cell line has been widely studied and its TER and cation permeability properties are well characterized. Cldn2 confers the leaky barrier properties of MDCK II cells. If Cldn2 is depleted, TER increases and sodium and chloride potentials are significantly reduced [31].

We found that quercetin differentially affected both the expression and the localization of some claudins over time. Quercetin also significantly increased transepithelial resistance. Our data suggest that after treatment with quercetin, the barrier becomes tighter, which could lead to a decrease in cation and water reabsorption. This may result in a more favorable urinary filtrate that is less prone to crystal supersaturation and the formation of a stone.

## 2. Results

### 2.1. Quercetin Increased Transepithelial Resistance of MDCK II Cells

Transepithelial resistance (TER) is a measure of the electrical resistance of a cell monolayer and is modulated by cell confluence, barrier permeability, and tight junction composition. For these studies, we used the cellZscope (NanoAnalytics) to continuously measure TER in MDCK II cells from the time of plating until several days after confluence when stable, mature tight junctions are formed. In cation-selective cell lines, TER increases as cells become closer or when paracellular movement of cations is reduced [30]. As predicted, there was an increase in TER immediately after seeding, when cells are dividing and confluence is increasing (Appendix A). Under control conditions, MDCK II cells reached a peak resistance of 130 Ω·cm^2^ ~40 h after seeding. Once maximum confluence is achieved, contact inhibition and tight junction remodeling takes place, which leads to a decrease in the TER that eventually stabilizes and becomes constant as the cells form stable mature tight junctions [32,33]. In MDCK II cells, this translated into a decrease in TER that then remained at 50–80 Ω·cm^2^, which is characteristic for mature tight junctions in this model [33]. Therefore, this was the time point selected for treatment with quercetin since it best represents the mature renal proximal tubule epithelium.

We confirmed that 400 µM quercetin, the concentration used in previous studies [21,22], was also the most optimal for our experiments through a dose response curve (Appendix A). We monitored the effects of quercetin on TER for ~96 h after treatment. Figure 1A shows the TER profile for one of the three biological replicates, which were each done in triplicate. Analysis of all data showed a small expected increase in TER immediately after treatment, due to the removal of the cells from the incubator and change of media. In control cells, the TER stabilized within 4–5 h and then remained at a resistance of ~50 Ω·cm^2^ (Figure 1A). Two-way ANOVA analysis showed that TER changed significantly over time and in a treatment-dependent manner (P_int_ = 0.0405; P_time_ < 0.0001; P_treat_ < 0.0001). Cells treated with 400 µM quercetin exhibited a progressive increase in TER, reaching significance at 3 h when the TER was ~15 Ω·cm^2^ higher than the control cells (*p* = 0.049; quercetin-treated = 90.04 ± 4.01 Ω·cm^2^ versus control cells = 70.7 ± 1.62 Ω·cm^2^). The TER remained significantly increased until 5 h post-treatment (*p* = 0.046; quercetin-treated = 86.33 ± 2.94 Ω·cm^2^ versus control cells = 66.86 ± 3.59 Ω·cm^2^) and then progressively decreased to ~5 Ω·cm^2^ below control levels 15 h after treatment, which was not significant (15 h: *p* > 0.99; control 60.08 ± 3.61, quercetin 62.21 ± 2.37). Following the decrease, TER increased again, reaching a steady state level of 10 Ω·cm^2^ above control, 36 h after treatment, which was statistically significant and remained significantly increased for the duration of the experiment (36 h: *p* = 0.0071; control 54.7 ± 2.31, quercetin 78.05 ± 5.19) (48 h: *p* < 0.0001; control 53.35 ± 1.8, quercetin 85.68 ± 2.55) (Figure 1).

### 2.2. Quercetin Treatment Caused Claudin-Specific Changes in Expression and Membrane Localization

To determine if the changes in TER caused by quercetin treatment corresponded with different claudin profiles, cells were collected 1, 6, 24, and 48 h after treatment. Claudin expression was assessed by western blot analysis and localization to the tight junction barrier was assessed by immunofluorescence. Immunofluorescence also provided a qualitative assessment of claudin expression. Five claudins expressed in MDCK II cells were studied: Cldn1, -3, and -7 that have barrier-sealing functions, Cldn2 that is involved in cation pore formation, and Cldn4 that is involved in anion pore formation. For all experiments, cells were cultured for 72 h before treatment with 400 µM quercetin to ensure that the cells had established mature tight junctions.

#### 2.2.1. Cldn1

Western blot analysis revealed a significant decrease in Cldn1 expression over time in both controls and quercetin-treated cells (P_time_ = 0.021). Quercetin treatment significantly lowered Cldn1 levels at 48 h compared to controls (*p* = 0.038; control 1.47 ± 0.55; quercetin 0.44 ± 0.18). A change in the relative abundance in the two migratory bands was observed at 24 h, although the total amount of Cldn1 was not affected (Figure 2A,B). Immunofluorescence analysis revealed decreased levels of Cldn1 at 1, 6, and 48 h (Figure 2C). A reduction in Cldn1 co-localization with ZO1 can be seen at 1 and 48 h, although it was not significant (*p* = 0.3 and *p* = 0.2, respectively) (Figure 2D). These data suggest than even though general levels of Cldn1 were decreased, the remaining Cldn1 still co-localized with ZO1.

#### 2.2.2. Cldn2

Two-way ANOVA analysis on Cldn2 expression showed that Cldn2 decreased significantly with quercetin treatment (P_treat_ = 0.0021), while changes in time and interaction only suggested significance (P_int_ = 0.0525; P_time_ = 0.096). Sidak’s multiple comparison test showed that the decrease observed at 48 h after treatment with quercetin was significant (*p* = 0.0011; control 2.0 ± 0.79; quercetin 0.06 ± 0.057). At 24 h, no Cldn2 was observed by western blot analysis in quercetin-treated cells (Figure 3A,B). As previously reported, Cldn2 detection by immunofluorescence was patchy within the epithelial monolayer of control MDCK II cells [34], with clusters of cells showing high levels of Cldn2 expression and adjacent groups of cells showing virtually no expression (Figure 3C). Localization of Cldn2 was significantly changed after treatment with quercetin (P_treat_ = 0.0055). At 1 h and 6 h after quercetin treatment, no changes in localization of Cldn2 were observed. However, at 24 h and 48 h after treatment, quercetin-treated cells showed a significant reduction both in the amount of Cldn2 and in the portion of Cldn2 that co-localized with ZO1 (24h: *p* = 0.036; control 0.54 ± 0.11, quercetin 0.14 ± 0.08; 48 h: *p* = 0.034; control 0.52 ± 0.12, quercetin 0.12 ± 0.05) (Figure 3C,D).

#### 2.2.3. Cldn3

Western blot analysis showed that quercetin caused a significant treatment-dependent increase in Cldn3 expression (P_treat_ = 0.0395). In marked contrast to the effects on Cldn1 and -2, Cldn3 expression was increased at 24 h and 48 h after treatment, although it was only statistically significant at 48 h (*p* = 0.04; control 0.29 ± 0.1; quercetin 2.4 ± 0.98) (Figure 4A,B). Although there appeared to be a general decrease in Cldn3 detection by immunofluorescence after 1 h and 6 h of treatment (Figure 4C), co-localization analysis showed that there was no effect on Cldn3 co-localization with ZO1 at 1 h and 6 h (*p* = 0.79 and *p* = 0.99, respectively) (Figure 4D). Cldn3 also co-localized with ZO1 at later time points.

#### 2.2.4. Cldn4

Cldn4 expression changed over time following quercetin treatment (P_int_ < 0.0001; P_time_ = 0.0017; P_treat_ < 0.0001). At 24 h and 48 h, Cldn4 expression was significantly increased by western blot analysis (control = 0.67 ± 0.37; quercetin = 2.74 ± 0.36, *p* < 0.0001 and control = 0.79 ± 0.36; quercetin = 2.93 ± 0.37, *p* < 0.0001, respectively) (Figure 5A,B). In contrast, by immunofluorescence, qualitatively, Cldn4 was downregulated at 48 h (Figure 5C), but co-localized with ZO1 (Figure 5D). This could be a consequence of possible claudin modifications that may avoid correct recognition of the epitope by the antibody. At this point, we cannot explain the discrepancy in Cldn4 expression as assessed by western blot and immunofluorescence.

#### 2.2.5. Cldn7

In contrast to the other claudin family members examined, Cldn7 expression and localization were not affected by quercetin treatment (Figure 6). At 24h, there was an increased level of Cldn7 immunofluorescence in the cytoplasm (Figure 6C) but this was not associated with increased co-localization with ZO1 at the tight junction (Figure 6D). This may suggest that Cldn7 incorporation into the tight junction complex is saturated at this time point.

### 2.3. Quercetin’s Effects on Claudin Expression and Localization is Independent of the PI3K/AKT Pathway

Quercetin has been identified as a strong inhibitor of the PI3K/AKT pathway [35,36]. Given that receptor tyrosine kinases and pAKT phosphorylate and regulate claudins both directly and indirectly [37,38], we hypothesized that quercetin affects claudin expression through the PI3K/AKT pathway. Western blot analysis of pPI3K, AKT, and pAKT was performed following quercetin treatment. The levels of pPI3K and AKT in MDCK II cells did not change significantly after quercetin treatment. pAKT was reduced in quercetin-treated cells at 1 h and 6 h after treatment. pAKT levels could not be detected at 1 h and 6 h after quercetin treatment limiting the statistical analysis at these time points. At 24 h and 48 h, pAKT was decreased, but this was not statistically significant (*p* = 0.99 and *p* = 0.7, respectively). These data suggest that quercetin inhibits the AKT/pAKT pathway in MDCK type II cells (Figure 7A,B).

To query if quercetin’s effects on claudin expression were due to impairment of the PI3K/AKT pathway, we inhibited or activated the PI3K/AKT pathways using wortmannin or SC79, respectively, alone or in combination with quercetin. As predicted, treatment with wortmannin caused a transient decrease in pAKT in MDCK II cells at 1 h and 6 h and SC79 increased pAKT at 1 h and 6 h (Figure 7C–F). In both cases, effects were greatly attenuated by 24 h. However, neither activation nor inhibition of the pAKT pathway recapitulated the effects on claudins observed following quercetin treatment (Figure 8), and neither wortmannin nor SC79 rescued the effects of quercetin. Therefore, our data suggest that quercetin is not acting through the PI3K/AKT pathway to effect changes in claudin protein expression.

## 3. Discussion

The purpose of this study was to determine the effect of quercetin on the cation-selective tight junction barrier in MDCK II cells, which models the proximal tubule of the kidney. We showed that the effects of quercetin on the MDCK barrier were stabilized 48 h after treatment, resulting in an increased TER of at least 20% over the TER observed in untreated cells. The quercetin-dependent increase in TER correlates with decreased expression of Cldn1 and decreased expression and localization of Cldn2 to the tight junction. Although a significant increase in Cldn3 and -4 was observed by western blot analysis, tight junction localization of Cldn3, -4, and -7 were not affected at this timepoint.

In the mature kidney, transepithelial resistance in the proximal tubule is <10 Ω·cm^2^, indicating that this nephron segment has a leaky barrier, which correlates with its important role in salt and water reabsorption [30]. Therefore, the increase of 10–20 Ω·cm^2^ observed following quercetin treatment of MDCK II cells would be very relevant in the context of the proximal tubule, where similar oscillations could completely change its barrier properties.

The TER oscillation observed during the first 48 h after quercetin treatment reflects dynamic tight junction remodeling and correlates with the changes in claudin expression that we observed. The decrease in TER observed at 6h coincided with a decrease in the immunofluorescent levels of barrier-sealing claudins, Cldn1 and -3. This is predicted to translate into a ‘leakier’ barrier, and consequently, a reduced TER. The increased TER at 24 h and 48 h coincided primarily with a decrease in the expression and tight junction localization of Cldn2, a cation-selective pore-forming claudin. Decreased Cldn2 at the tight junction is predicted to reduce paracellular movement of cations [31]—including sodium and calcium, and water—and may contribute to the increase in TER, as previously described for Cldn4 in MDCK [30].

Our data and those obtained by other groups, suggest that the effects of quercetin are cell-line dependent [21,22,23,24,31,39]. When Tokuda and Furuse depleted Cldn2 from MDCK II cells, they observed increased TER and decreased sodium and chloride potential [31]. The increase in TER was significantly higher than what we observed (>1000 Ω cm^2^), perhaps due to compensation by other claudins in response to removal of Cldn2. In their experiment, depletion of Cldn2 led to increased tight junction localization of Cldn1, -3, -4, and -7, which would effectively tighten the barrier and increase TER. In our quercetin-treated cells, the 90% reduction of Cldn2, was not accompanied by an increased localization of other claudins to the tight junction: Cldn1 expression was decreased, Cldn3 and -4 expression was increased but localization to the tight junction was unchanged, and Cldn7 was unchanged. Thus, quercetin causes a different net effect on claudin expression/localization compared to depletion of Cldn2 alone and, as a result, there is a more modest increase in TER.

Quercetin also increased TER of the anion-selective barrier in LLC-PK1 cells, here, expression of Cldn2 and -3 were downregulated while Cldn4, -5, and -7 were upregulated [21]. However, claudin localization was not assessed. An oscillation and final increase in TER following quercetin treatment, has been observed in Caco-2 cells [23,24], where TER oscillated similarly to what we observed in MDCK II cells during the first 48 h after treatment. In this case Cldn1 was displaced to the cytoskeletal fraction, Cldn4 expression was increased, while Cldn3 was unperturbed. In contrast, Valenzano et al., did not report any effect on TER after 17 h of quercetin treatment on Caco-2 cells, although Cldn2, -4, and -5 were increased [22]. These data together with our findings indicate that quercetin is able to tighten tight junction barriers, but does so by differentially affecting different claudin family members, in a cell-type specific manner.

We observed two migratory bands for Cldn1 and -4 in our western blot analyses. Previous studies have linked these different migratory bands to either claudin degradation [40] or claudin phosphorylation [41,42]. Claudins are known to undergo post-translational modifications, including phosphorylation, glycosylation, palmitoylation, and ubiquitination [43]. These modifications are thought to rapidly change claudin stability at the tight junction. For instance, dephosphorylation of Cldn1 and -2 is a signal for degradation and reduces their presence at tight junctions [44,45]. In the case of Cldn4, studies have shown that the effects depend on the site of phosphorylation. In some cases, phosphorylation leads to the disruption of the tight junction [46,47], while in others, phosphorylation increases the stability of Cldn4 at the tight junction [41]. Further experiments are required to elucidate if the two migratory bands are a consequence of different posttranslational modifications, or if they correspond to degradation products [40].

Other studies that have looked at quercetin’s effects on tight junction barrier properties and claudin expression have been based only on western blot analyses. The lack of congruence between our western blot and immunofluorescence expression data for some of the claudins may reflect the availability of claudin epitopes for detection using these two methods. It could also reflect disparities in how posttranslational modifications or protein degradation can be assessed by western blot versus immunofluorescence. For instance, in the case of Cldn1, at 24 h and 48 h, the western blot data shows a loss of the slower migrating band and decreased Cldn1 expression in the quercetin-treated cells. In contrast, the levels by immunofluorescence are equivalent to the control cells. This could suggest that the immunofluorescence signal is primarily due to recognition of the faster migrating band.

Cldn2 immunofluorescence exhibited quite a different pattern compared to other claudin family members. In contrast to most claudins, Cldn2 was highly expressed in some regions and almost absent in other regions within confluent cell layer. This has been previously described and high Cldn2 levels appear to correlate with increased cell confluence [34]. The increased Cldn2 during tight junction maturation, can be clearly seen in the control cells by both western blot analysis and by colocalization with ZO1 at tight junctions.

The mechanism(s) by which quercetin impacts the tight junction barrier through its effects on claudin expression and localization remains unresolved. Quercetin is known to bind to several membrane-bound nutraceutical receptors that signal through different pathways to reduce oxidative stress. Quercetin also interacts directly with several kinases [48,49,50,51,52]. Some studies show that it is a potent inhibitor of PI3K/AKT pathway [36,53]; although there is some controversy [49,54,55], while others show that it inhibits the ERK and NF-κβ pathways [56], and activates the AMPK pathway [49,57]. However, these studies have been done on different cell lines, using different conditions and measuring different outcomes. Therefore, it remains unclear if quercetin acts upstream of all of these pathways, or if its effects are cell-type specific. We showed that quercetin inhibits PI3K/AKT pathway in MCDK II cells. However, inhibition the PI3K/AKT pathway was not sufficient to recapitulate quercetin’s effects on claudin expression.

So, how does this translate to the potential therapeutic effects of quercetin to prevent kidney stone formation in the context of the proximal tubule? The site of kidney stone formation is not well understood. Some studies suggest that kidney stones are formed in the interstitium, while others suggest that they form in the lumen of the nephron [2,58,59]. Here, we showed that quercetin led to a relative increase in barrier claudins at the tight junction relative to the pore-forming claudins in MDCK II cells. Although direct measurements of cation potential and water permeability were not performed, our data suggest that quercetin may be beneficial by preventing sodium, calcium, and water within the urine from crossing the epithelial barrier to enter the interstitium. We believe that the effects seen for Cldn2 are essential to explain what may be happening in the nephron. Cldn2 is a cation-selective pore forming claudin that promotes the paracellular movement of sodium, calcium, and water [29]. A decrease in Cldn2 is predicted to lead to a reduction in reabsorption of these substances. This would then prevent the transport of sodium and calcium in the interstitium which can drive stone formation. Alternatively, quercetin may effectively tighten the proximal tubular epithelium to maintain the water content within the urinary lumen of this segment to keep calcium and sodium solubilized [60]. This mechanism could correlate with the beneficial effects that high-water diets or diuretic drugs have in the prevention of kidney stones. In vivo experiments are necessary to discern which specific scenario is taking place and whether this effect could be beneficial for the prevention of kidney stones.

## 4. Material and Methods

### 4.1. Cell Culture

Madin–Darby canine kidney cells II (MDCK II) cells were obtained from ATCC. They were incubated at 37 °C and 5% CO2 in Dulbecco’s modified Eagle’s medium (DMEM) supplemented with 10% FBS, 51 IU penicillin, 50 µg/mL streptomycin and 16 µg/mL of gentamicin (Wisent BioProducts, Quebec, QC, Canada).

### 4.2. Chemicals and Antibodies

Quercetin (Sigma-Aldrich, Darmstadt, Germany) was added to the culture media, and heated at 37 °C with continuous stirring for at least 30 min in order to ensure that it was dissolved. It was used at the working concentration of 400 μM. PI3K/AKT pathway modulators Wortmannin (Abcam, 120148, Cambridge, UK) and SC79 (Abcam, 146428, Cambridge, UK) were used at working concentrations of 2 μM and 22 nM respectively.

Primary antibodies used for immunofluorescence and western blotting were: Cldn1 (Invitrogen, 374900, Carlsbad, CA, USA), Cldn2 (Invitrogen, 516100, Carlsbad, CA, USA), Cldn3 (Abcam, 15102, Cambridge, UK), Cldn4 (Invitrogen, 364800, Carlsbad, CA, USA), Cldn7 (Spring Bioscience, E10594, Pleasanton, CA, USA), ZO1 (Invitrogen, 339100, Carlsbad, CA, USA), pPI3K (Cell signaling, 4228, Danvers, MA, USA), pAKT (Cell signaling, 9271, Danvers, MA, USA), AKT (Cell signaling, 9272, Danvers, MA, USA), and pan-actin (Cell signaling, 4968, Danvers, MA, USA,). In addition, secondary goat anti-rabbit (Alexa Fluor 595 and 488, Invitrogen, Carlsbad, CA, USA), goat anti mouse (Alexa Fluor 595 and 488, Invitrogen, Carlsbad, CA, USA), goat anti-rabbit-HRT conjugated (Cell Signaling, 70748, Danvers, MA, USA), and goat anti mouse peroxidase conjugated (Jackson ImmunoResearch, 115-035-146, West Grove, PA, USA) were used.

### 4.3. Transepithelial Resistance (TER)

Cells were seeded on sterile 0.4 µm pore size, 12-well transparent polyethylene terephthalate inserts (Corning, NY, USA) at a cell density of 0.15 × 10^6^ cells/insert and placed in the automated cell monitoring CellZscope^®^ system (NanoAnalytics, Münster, Germany). The CellZscope^®^ system provides noninvasive, continuous monitoring of cell monolayers impedance, capacitance, and resistance.

The tissue culture inserts were placed into a 12-well cell module, and the system was incubated at 37 °C and 5% CO2. TER measurements, expressed in ohm square centimeters, were recorded in real time every 20 min. Insert background resistance was automatically subtracted by the CellZscope^®^ system. DMEM in both the apical and basal compartments was replaced with fresh DMEM (control) or DMEM plus 400 μM quercetin at 72 h, at which point mature tight junctions were established and stabilized (Resistance ≅ 70 Ω·cm^2^). Measurements were recorded in real time every 20 min for at least 90 h following treatment. Each time the machine was removed from the incubator or the media was changed, a small increase in TER is expected which then stabilizes.

### 4.4. Immunofluorescence Staining

Cells were seeded on coverslips in 12-well plates at a cell density of 0.1 × 10^6^ cells/well. Once confluence and TJ maturity were achieved (≈ 72 h), cells were treated with DMEM (control) or DMEM plus 400 μM quercetin and collected at 1, 6, 24, or 48 h. Cell layers were rinsed with phosphate buffer saline (PBS) and fixed in 10% trichloroacetic acid for 15 min at 4 °C. Cells were then blocked with 10% normal goat serum in phosphate buffer saline and 0.3% Triton-100 (PBST) and incubated overnight at 4 °C with 5% normal goat serum in PBST and primary antibodies: Cldn1, -2, -3, and -4 (1:100), Cldn7 (1:50), and ZO1 (1:150). Cells were washed with PBS. Alexa Fluor-conjugated secondary antibodies (1:500) in PBST were added for 1 h at RT. Coverslips were washed with PBS and then placed on slides with Slowfade Gold with DAPI (Invitrogen, Carlsbad, CA, USA). Z-stacks were imaged using a Zeiss LSM780 laser scanning confocal microscope. To determine the amount of claudin localized in the membrane, maximum intensity projections were obtained for each stack of images and then ZEN software colocalization analysis was performed. Only M2 results are presented; M2 makes reference to how many of the red pixels (ZO1) coincide with green pixels (claudin). A high M2 indicates high claudin at the membrane while a low M2 indicates high ZO1 alone, which relates to a low claudin at the membrane. M2 is not a reading of abundance or intensity levels and therefore it only provides information about localization.

### 4.5. Western Blot Analysis

Cells were seeded in 10 cm plates at a cell density of 2 × 10^6^ cells/plate. Once confluence and TJ maturity was reached (≈ 72 h), cells were treated with DMEM (control) or DMEM plus 400 μM quercetin for 1, 6, 24, or 48 h, and then collected by physical scraping into lysis buffer (25 mM Tris-HCl Ph7.4; 10 mM sodium pyrophosphate; 25 mM NaCl; 10 mM sodium fluoride; 2 mM EGTA; 2 mM EDTA; 1% NP40; 0.1% SDS; 1.45 nM Pepstatin A; 2.1 nM Leupeptin; 0.15 nM Aprotinin, and 0.57 μM PMSF). Protein concentration was determined by Bradford assay and 50 μg per sample were separated by 12.5% SDS-PAGE and transferred onto PVDF membranes. Membranes were blocked for 1 h with 5% non-fat milk in PBST or with 5% BSA in Tris-NaCl buffer and 0.3% of Tween-20 (TBST). Membranes were incubated overnight at 4 °C with primary antibodies in PBST or TBST: Cldn1, -2, -3, -4, -7, pPI3K, pAKT, and AKT (1:1000) or pan-actin (1:2000). Membranes were then washed with PBST or TBST and then incubated with goat-anti-rabbit-HRT conjugated or goat-anti-mouse-peroxidase conjugated (1:5000) secondary antibodies for 1h at RT. Membranes were revealed using Clarity Western ECL substrate (Biorad laboratories, 1705061, Hercules, CA, USA), and imaged using Amersham imager 600 (GE Healthcare, Little Chalfont, United Kingdom). Band densities were quantified by densitometry using ImageJ, normalized against the 1 h control sample on each blot and then to normalized actin. Because some variability was observed in actin, we performed a statistical analysis of actin expression from all experiments and determined that there was no significant change over time or in response to quercetin treatment.

### 4.6. Statistics

All plotted results are mean ± SEM. Statistical analyses were performed using Graph-Pad PRISM (version 8.02, GraphPad Software, Inc., California, USA). Two-way ANOVA analysis was used for effects of treatment, time, and interaction. Sidak’s multiple comparison test was used to compare the effect of treatment at the different time points. For TER Appendix A, unpaired T-test were performed at the specific time points. Significance is considered if *p* < 0.05.

## Figures and Tables

**Figure 1 ijms-20-04889-f001:**
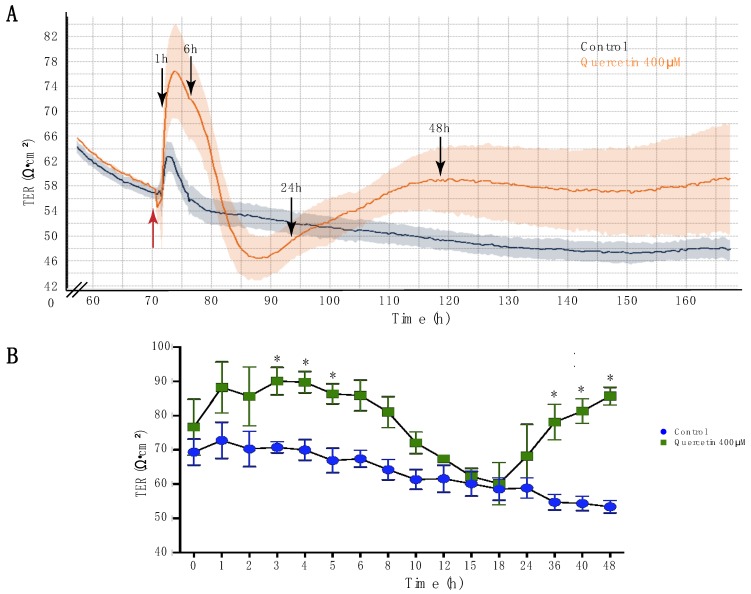
Quercetin caused oscillations in transepithelial resistance (TER) of MDCK II cells. (**A**) Representative plot of TER in control cells (black) and cells treated with 400 µM quercetin (orange) from one biological replicate performed in triplicate. Red arrow indicates when quercetin was added to the culture medium. Black arrows indicate the time points taken for western blot and immunofluorescence analysis. (**B**) TER of control and quercetin-treated cells at different time points after treatment from three independent experiments performed in triplicate. Two-way ANOVA was performed (P_int_ = 0.04; P_time_ < 0.0001; P_treat_ < 0.0001). Mean and SEM are plotted. * Denotes significance, *p* < 0.05.

**Figure 2 ijms-20-04889-f002:**
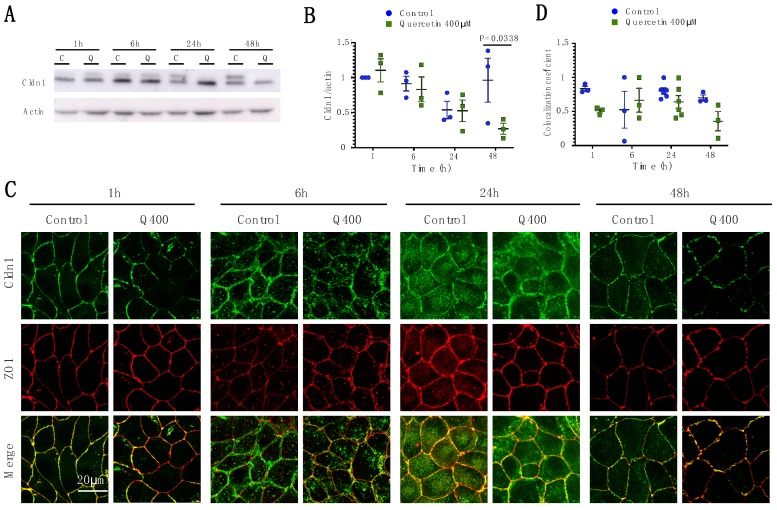
Analysis of Cldn1 expression and localization in MDCK II cells following quercetin treatment. (**A**) Western blot analysis of Cldn1 expression in cell lysates from control and 400 µM quercetin-treated MDCK II cells. Actin was used as a loading control. (**B**) Densitometry measurements of Cldn1 and actin intensity on western blot were normalized to expression at 1h. Normalized Cldn1 expression relative to normalized actin expression is plotted (P_int_ = 0.11; P_time_ = 0.022; P_treat_ = 0.16). (**C**) Immunofluorescence of control and 400 µM of quercetin-treated MDCK II cells at 1, 6, 24, and 48 h. Cldn1 is shown in green and ZO1 is shown in red. (**D**) Localization of claudin at the tight junction was assessed by determining the amount of ZO1 that was co-localized with claudin expression (P_int_ = 0.24; P_time_ = 0.35; P_treat_ = 0.055). For all graphs, each point corresponds to an independent experiment; mean and SEM are shown. C = Control and Q = Quercetin. Scale bar, 20 µm.

**Figure 3 ijms-20-04889-f003:**
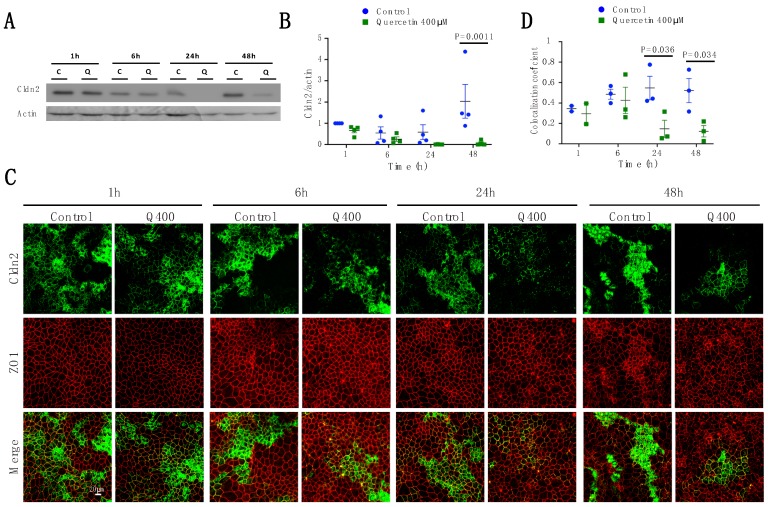
Analysis of Cldn2 expression and localization in MDCK II cells following quercetin treatment. (**A**) Western blot analysis of Cldn2 expression in cell lysates from control and 400 µM quercetin-treated MDCK II cells. Actin was used as a loading control. (**B**) Densitometry measurements of Cldn2 and actin intensity on western blot were normalized to control at 1 h. Normalized Cldn2 expression relative to normalized actin is plotted (P_int_ = 0.052; P_time_ = 0.096; P_treat_ = 0.0021). (**C**) Immunofluorescence of control and 400 µM of quercetin-treated MDCK II cells at 1, 6, 24, and 48 h. Cldn2 is shown in green and ZO1 is shown in red. (**D**) Localization of claudin at the tight junction was assessed by determining the amount of ZO1 that co-localized with claudin expression (P_int_ = 0.15; P_time_ = 0.46; P_treat_ = 0.0055). For all graphs, each point corresponds to an independent experiment; mean and SEM are shown. C = Control and Q = Quercetin. Scale bar, 20 µm.

**Figure 4 ijms-20-04889-f004:**
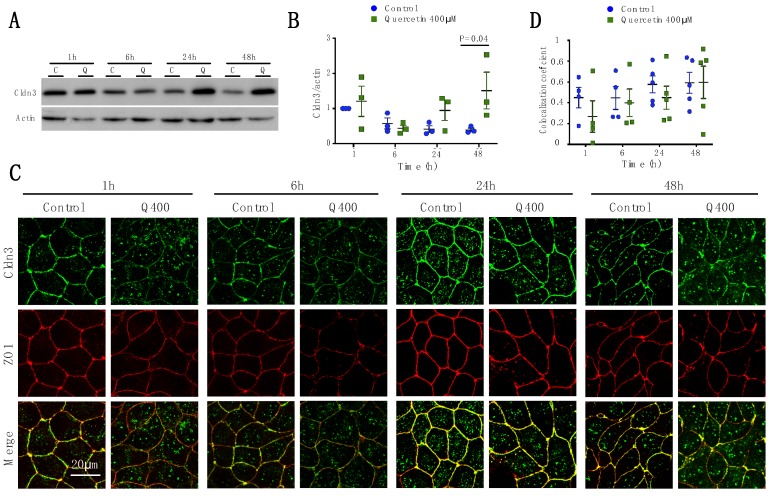
Analysis of Cldn3 expression and localization in MDCK II cells following quercetin treatment. (**A**) Western blot analysis of Cldn3 expression in cell lysates from control and 400 µM quercetin-treated MDCK II cells. Actin was used as a loading control. (**B**) Densitometry measurements of Cldn3 and actin were normalized to control at 1 h. Normalized Cldn3 expression relative to normalized actin is plotted (P_int_ = 0.16; P_time_ = 0.16; P_treat_ = 0.039). (**C**) Immunofluorescence of control and 400 µM of quercetin-treated MDCK II cells at 1, 6, 24, and 48 h. Cldn3 is shown in green and ZO1 is shown in red. (**D**) Localization of claudin at the tight junction was assessed by determining the amount of ZO1 that co-localized with claudin expression (P_int_ = 0.87; P_time_ = 0.24; P_treat_ = 0.31). For all graphs each point corresponds to an independent experiment; mean and SEM are shown. C = Control and Q = Quercetin. Scale bar, 20 µm.

**Figure 5 ijms-20-04889-f005:**
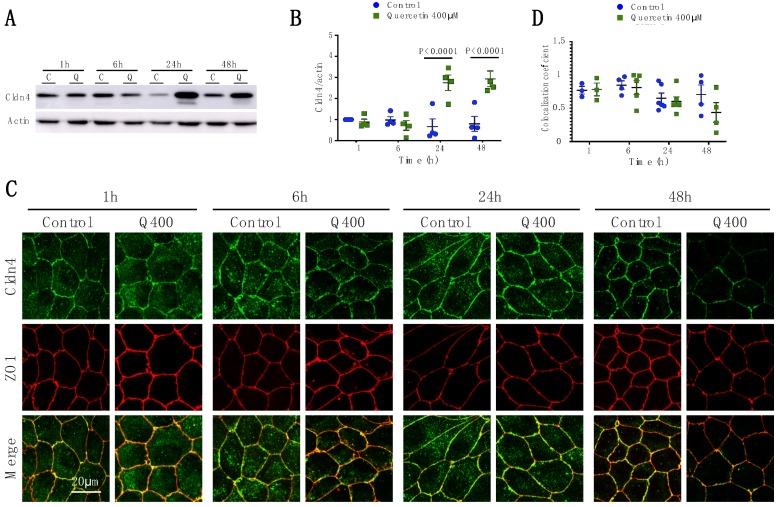
Analysis of Cldn4 expression and localization in MDCK II cells following quercetin treatment. (**A**) Western blot analysis of Cldn4 expression in cell lysates from control and 400 µM quercetin-treated MDCK II cells. Actin was used as a loading control. (**B**) Densitometry measurements of Cldn4 and actin intensity on western blot were normalized to control levels at 1 h. Normalized Cldn4 expression relative to normalized actin is plotted (P_int_ < 0.0001; P_time_ = 0.0017; P_treat_ < 0.0001). (**C**) Immunofluorescence of control and 400 µM of quercetin-treated MDCK II cells at 1, 6, 24, and 48 h. Cldn4 is shown in green and ZO1 is shown in red. (**D**) Localization of claudin at the tight junction was assessed by determining the amount of ZO1 that co-localized with claudin expression (P_int_ = 0.54; P_time_ = 0.051; P_treat_ = 0.24). For all graphs, each point corresponds to an independent experiment; mean and SEM are shown. C = Control and Q = Quercetin. Scale bar represent 20 µm.

**Figure 6 ijms-20-04889-f006:**
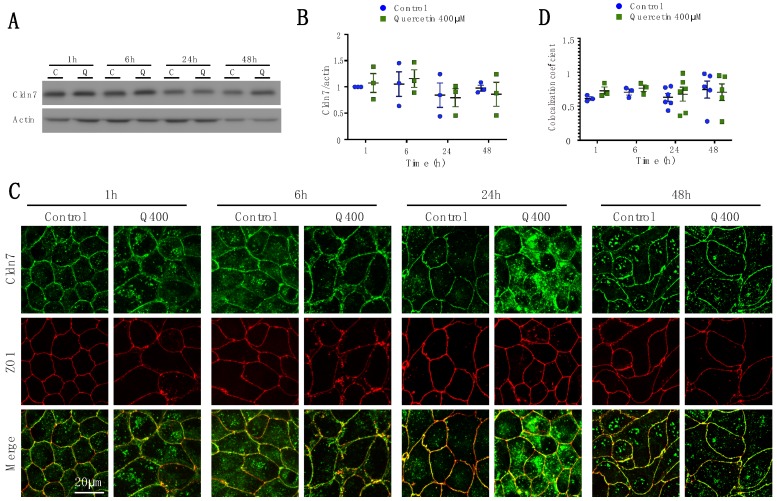
Analysis of Cldn7 expression and localization in MDCK II cells following quercetin treatment. (**A**) Western blot analysis of Cldn7 expression in cell lysates from control and 400 µM quercetin-treated MDCK II cells. Actin was used as a loading control. (**B**) Densitometry measurements of Cldn7 and actin intensity on western blot were normalized to control at 1h. Normalized Cldn7 expression relative to normalized actin is plotted (P_int_ = 0.916; P_time_ = 0.41; P_treat_ = 0.98). (**C**) Immunofluorescence of control and 400 µM of quercetin-treated MDCK II cells at 1, 6, 24, and 48 h. Cldn7 is shown in green and ZO1 is shown in red. (**D**) Localization of claudin at the tight junction was assessed by determining the amount of ZO1 that co-localized with claudin expression (P_int_ = 0.89; P_time_ = 0.8; P_treat_ = 0.55). For all graphs each point corresponds to an independent experiment; mean and SEM are shown. C = Control and Q = Quercetin. Scale bar, 20 µm.

**Figure 7 ijms-20-04889-f007:**
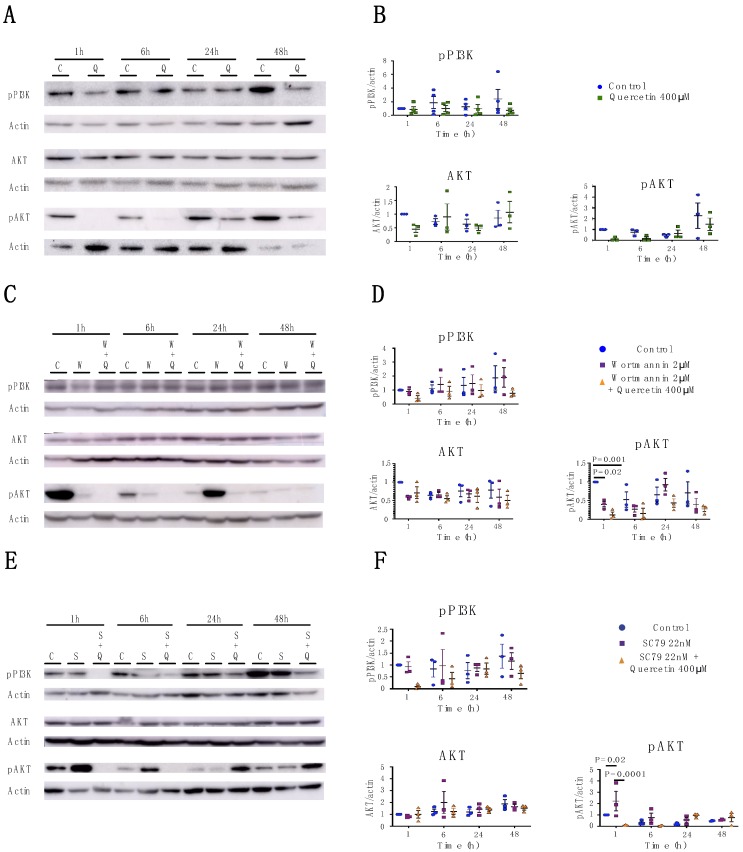
Analysis of pPI3K, pAKT, and AKT expression in MDCK II cells following quercetin treatment. (**A**) Western blot analysis of claudin expression in cell lysates from control and 400 µM quercetin-treated MDCK II cells. Actin was used as a loading control. (**B**) Densitometry measurements of band intensity on western blot were normalized to control at 1 h and protein expression was plotted relative to normalized actin expression. pPI3K (P_int_ = 0.67; P_time_ = 0.79; P_treat_ = 0.14), AKT (P_int_ = 0.44; P_time_ = 0.49; P_treat_ = 0.68), and pAKT (P_int_ = 0.68; P_time_ = 0.027; P_treat_ = 0.142) measurements were plotted. (**C**) Western blot analysis of pPI3K, pAKT, and AKT expression in cell lysates from control, 2 µM wortmannin-treated and 2 µM wortmannin plus 400 µM quercetin-treated MDCK II cells. Actin was used as a loading control. (**D**) Densitometry measurements of band intensity on western blot were normalized to control at 1 h and protein expression was plotted relative to normalized actin expression. pPI3K (P_int_ = 0.97; P_time_ = 0.03; P_treat_ = 0.13), AKT (P_int_ = 0.79; P_time_ = 0.53; P_treat_ = 0.099), and pAKT (P_int_ = 0.15; P_time_ = 0.064; P_treat_ = 0.0008) measurements were plotted. (**E**) Western blot analysis of pPI3K, pAKT, and AKT expression in cell lysates from control 22 nM SC79-treated and 22 nM SC79 plus 400 µM quercetin-treated MDCK II cells. Actin was used as a loading control. (**F**) Densitometry measurements of band intensity on western blot were normalized to control at 1 h and protein expression was plotted relative to normalized actin expression. pPI3K (P_int_ = 0.81; P_time_ = 0.51; P_treat_ = 0.078), AKT (P_int_ = 0.79; P_time_ = 0.52; P_treat_ = 0.099), and pAKT (P_int_ = 0.16; P_time_ = 0.063; P_treat_ = 0.0008) measurements were plotted. Each point corresponds to an independent experiment; mean and SEM are shown. W = Wortmannin; S = SC79; C = Control; and Q = Quercetin.

**Figure 8 ijms-20-04889-f008:**
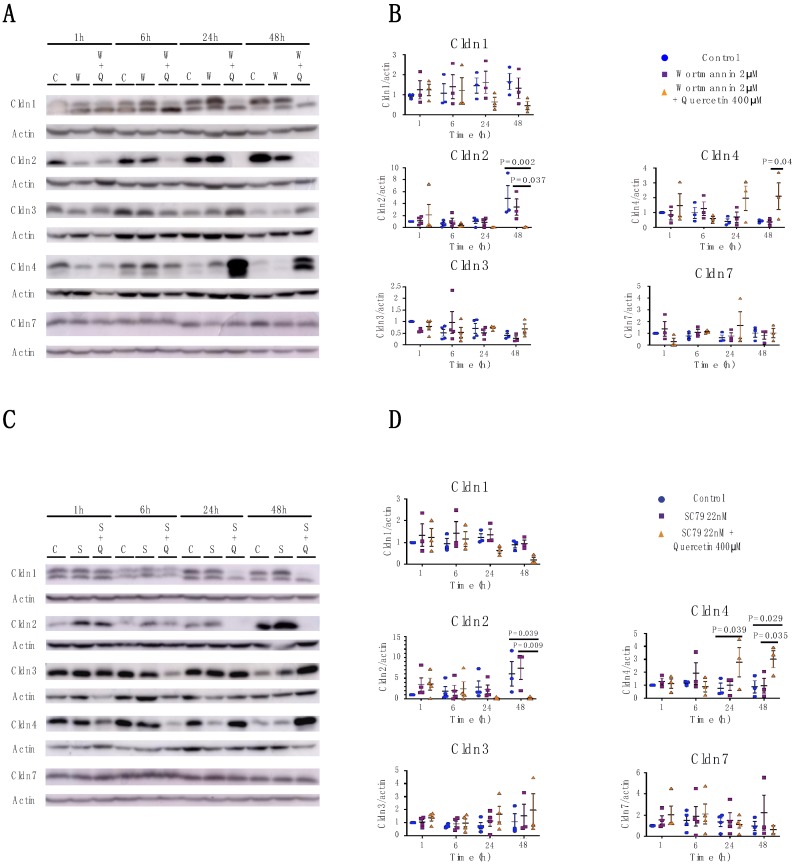
Analysis of Cldn1, -2, -3, -4, and -7 expression in MDCK II cells following wortmannin or SC79 treatment. (**A**) Western blot analysis of claudin expression in cell lysates from control, 2 µM wortmannin-treated and 2 µM wortmannin plus 400 µM quercetin-treated MDCK II cells. Actin was used as a loading control. (**B**) Densitometry measurements of band intensity on western blot were normalized to control at 1 h and protein expression was plotted relative to normalized actin expression. Cldn1 (P_int_ = 0.6; P_time_ = 0.98; P_treat_ = 0.24), Cldn2 (P_int_ = 0.059; P_time_ = 0.016; P_treat_ = 0.097), Cldn3 (P_int_ = 0.29; P_time_ = 0.26; P_treat_ = 0.78), Cldn4 (P_int_ = 0.21; P_time_ = 0.97; P_treat_ = 0.041), and Cldn7 (P_int_ = 0.41; P_time_ = 0.98; P_treat_ = 0.83) measurements were plotted. (**C**) Western blot analysis of claudin expression in cell lysates from control 22 nM SC79-treated and 22 nM SC79 plus 400 µM quercetin-treated MDCK II cells. Actin was used as a loading control. (**D**) Densitometry measurements of band intensity on western blot were normalized to control at 1 h and protein expression was plotted relative to normalized actin expression. Cldn1 (P_int_ = 0.6; P_time_ = 0.15; P_treat_ = 0.11), Cldn2 (P_int_ = 0.09; P_time_ = 0.14; P_treat_ = 0.11), Cldn3 (P_int_ = 0.99; P_time_ = 0.46; P_treat_ = 0.21), Cldn4 (P_int_ = 0.082; P_time_ = 0.73; P_treat_ = 0.047), and Cldn7 (P_int_ = 0.87; P_time_ = 0.78; P_treat_ = 0.5) measurements were plotted. Each point corresponds to an independent experiment; mean and SEM are shown. W = Wortmannin; S = SC79; C = Control; and Q = Quercetin.

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
