# Peer review of "Temporal Effects of Quercetin on Tight Junction Barrier Properties and Claudin Expression and Localization in MDCK II Cells"

_ijms, 2019, doi:10.3390/ijms20194889_

Round 1

Reviewer 1 Report

Major concerns:

Lack of proper statistical analysis needed to support conclusions of the study is evident. Actin blots that clearly show time-dependent and blot-dependent changes – so, should actin really be normalized to? If actin is to be used as a temporal control, its abundance in WB needs to be analyzed by a two-way ANOVA (factor 1 – time; factor 2 – treatment). Same for ZO-1 in IHC figures. Lastly, to back up time-dependent statements like “Cldn-X decreased in this group at Y hours post-treatment”, a two-way ANOVA is needed for data analysis, where independent factors are time and treatment with quercetin. Only in this case, an interaction term of the two-way ANOVA can be interpreted as a difference due to quercetin treatment – this will be necessary to demonstrate that the time-dependent changes in the untreated control samples are not masking what is happening in quercetin-treated samples TER used as proxy for permeability – is there no way to treat insert-cultured epithelia with quercetin directly in the inserts and measure paracellular tracer flux? Additionally, WB and IHC have both been performed on insert-grown epithelia to date in many systems, so methodology shouldn’t be a problem. No ion permeability measured – 15 Ohm difference may be statistically significant, but is it enough to alter cation permeability? Again, ion permeability has been measured in semi-permeable terephthalate inserts in many models to date. Data representation needs to be assessed, considering n=3 (Section 4.6) and n=2 (Fig 7), I would not recommend making quantitative arguments with a statistical power of n<5. Numbers of technical and biological replicates need to be explicitly stated in every graph for every averaged data point.

Detailed comments:

Introduction

43 – I think you mean “tetraspanin”; Tetraspan® is a registered trademark

l.44 – I believe the total number of claudins is close to 30 by now (Please double-check with the most recent review paper or the NCBI)

49-61 - Nothing mentioned in the intro on why cldn-1, -3 and -4 are investigated. How are barrier-sealing cldns involved in controlling cation reabsorption in the proximal segment of the nephron? As is, just sounds like you’re piggy-backing them to the study just because they are expressed in the model. Please add relevant info to the intro.

75-76 – the first statement in this paragraph needs backing of time-dependent statistics. None of the data in this study are analyzed with time taken into account - only fold-changes in comparison to control-treated inserts are reported at every time point. Please remedy or rewrite the intro and discussion to accurately reflect the data analysis.

77-78 –No measurements of ion or water permeability to back this up. Mostly conjecture – either make clear here that these parameters were not measured or remove this statement altogether, please.

Results

87-88 – “dramatic” is an overstatement - I wouldn’t call a 50-Ohm increase in TER “dramatic”

91-92 - So, the cells “plateau” and enter steady-state - why were the experiments not carried out at this stage (~130 hrs), but rather inserts were exposed to quercetin, while the TER was still decreasing (~70 hrs)? I wonder if the stats were performed on 70 hrs vs 130 hrs in controls, a time-dependent decrease in the TER would be apparent.

99-100 - How significant is a 15 Ohm*cm2increase in TER in relation to cation permeability, which is how the data re discussed in this study? Can you cite any data from previous studies that would suggest that this is enough to alter cation permeability? Or perform cation permeability yourself?

Fig 1A – why was the “undershoot” in TER not investigated in quercetin-treated inserts? Given that the long-term increase in the TER is the main focus of the paper, it seems significant to me that treatment with quercetin transiently decreased TER at ~14 h post-treatment.

122 – seems like the 24 hrs timepoint was used because it looked better mathematically -

why not investigate the undershoot at 14 h post-treatment instead?

ll.133-135 – any stats to back up these statements about time-dependent changes in Cldn 1 levels?  

Fig 2C - seemingly, ZO-1 levels change as well in control-treated inserts with time. Any comment on this?

149-150 – this doesn’t agree with Fig 3 - your blots actually show a decrease in Cldn-2 signal…

Fig 3C - Patchy Cldn-2 expression - is this typical of MDCK cells? Are you sure this is not lack of depth in confocal stacks used? Any citations to back up this expression pattern as normal in MDCK cells? Perhaps something to discuss if you think Cldn-2 abundance is important for your conclusions – it doesn’t appear that the whole epithelium is expressing it. Also, why was low-mag as in Fig 3 not used for quantification for all cldns (Figs 2, 4-6)? (if high-mag regions of interest are used selectively from Fig 2 images, the conclusions would be pretty different).

l.170 - Control levels of Cldn-3 likely affects this fold-change expression… If the data is expressed normalized to controls at every time point, decreasing control levels will result in a false positive increase in the normalized levels observed in the control group. This needs to be addressed with proper stats.

Fig 4A - Again, actin b/w C and Q at 1h seems different; same for Fig 4C – ZO-1 immunofluorescence clearly differs between control and treated groups at 1 h, which is probably inflating your results in Fig 4D. I agree that the qualitativetrend is obvious from the Cldn-3 staining pattern, but in order to back up the quantitative statements, proper time-dependent stats need to be run on normalization controls (actin and ZO-1) used to normalized data.

185 – “remained 4-fold elevated at 48 h” - You can’t say this unless you have longitudinal time-dependent statistics that back up this statement where time is one of the variables…

Fig 5A – Cldn-4 seems to be decreasing at 24h in controls – this may be amplifying your fold difference in treated samples; also, was double band in Q-treated at 24-48h addressed in the discussion?

Fig 6A – is actin at 48h lower in abundance? Why? If this is the case, all time-dependent statements must be removed from the paper – how can one justify a statement that abundance of something remains elevated or decreases after 48h, if normalization and fold controls themselves change?

222 – “reduced the levels of AKT at 1h” – this is not apparent from the WB included in Fig 7A

Fig 7 – this major issue applies to all WB figures - I am confused by the variability in actin signal at the same time point: does this suggest issues with blot transfer? (e.g., actin and PI3K are changing in opposite directions, which may be amplifying the fold-difference)

Have you done stats on actin alone to show that it is not significantly different between treatment groups? (a two-way ANOVA would be appropriate here) Also, three actin blots presented in this figure show completely different changes in abundance of actin over time, so is actin really a good “housekeeping” or “normalizing” control? Are you better off expressing everything relative to total protein? The data should really be reanalyzed as well - from your blots it seems that pI3K abundance is lower after treatment with Q at 1h - this is not apparent in Fig 7B. Likewise, quantifying pAKT signal using densitometry in Q-treated 1h group is inappropriate - there is no visible signal, therefore a quantitative statement may not be made! In short, methodology for data analysis and visualization needs to be reconsidered - WB are not representative of changes illustrated by bar graphs and normalization to actin seems inappropriate to me in case where it changes with time and is quite variable experiment-to-experiment.

235-237 – data not shown - a supplementary figure perhaps? Seems like these data are crucial for your argument.

Fig 7 – l.228 – how can one perform stats on an n=2??

Discussion

ll.245-246 - Not apparent to the reader how this would affect ion permeability. If there are studies examining ion permeability in relation to TER in this model and relating it to abundance of certain cldns, maybe cite them?

256- 257 - This “decrease” in TER after 6h does not seem significant to me, if it is, please reanalyze your data in a way that demonstrates this statistically. I also do not see the

Cldn-1 and Cldn-3 decrease in TJ localization in a way your data are analyzed and presented.

Cldn-1 and Cldn-3 do not exhibit decreased TJ localization in comparison to their controls at 6h.

Moreover, the data are not analyzed for longitudinal (temporal) changes, so 6h cannot be compared to earlier time points.

261 - Is there not a way to directly measure Na+, Ca2+or water permeability in this cell culture model? It has certainly been done in other cultured models.

273-275 - Perhaps, a phosphatase treatment to back this up? This is easily done in WB samples by incubating them with phosphatase prior to electrophoresis.

284-285 - Could you not add controls that would account for this? Peptide blocks?

304-306 – colocalization analysis is problematic as described above.

312 -313 - did not analyze relative cldn abundance – one cldn relative to another that is – no data to back up this statement

319-320 – this is largely conjecture – no ion or water permeability measurements were conducted.

322 – remove full stop before [52].

Materials and methods

331 – it is “Dulbecco” I believe 332 – 1% pen-strep is not informative – how much is this in molar or ug/ml?

l.339 – report all concentrations in the same units – molar or ug/ml

Section 4.3 - Was background resistance of an empty insert subtracted from your TER measurements?

Section 4.4 - Were samples from all time points processed for IHC together – same solutions, same incubation time? Again, just commenting on the variability in ZO-1 staining, which is used for co-localization measurements.

Section 4.5 - Some of the research suggests that epithelia grown on permeable inserts vs. solid support differ in TJ expression and barrier properties? Is there a reason why IHC and WB were not performed on cell culture inserts – this has certainly been done in other models.

Section 4.6 – this may be up to the editor, but in most cases three biological replicates do not constitute enough of for a quantitative argument – raw data should be included (individual values, average +/- s.e.m. as graphed in figures and p-values of statistical tests supplied. N numbers should be indicated in every figure individually – “at least three biological replicates” is not informative enough. The numbers of technical and biological replicates should be stated in each case. Technical and biological replicates should not be pooled together for statistical analysis to avoid pseudoreplication.

Reviewer 2 Report

The manuscript "Temporal effects of quercetin on tight junction barrier properties and claudin expression and localization in MDCK II cells" by Gamero-Estevez et al., analyzed a temporal response of the five major claudins in MDCK II cells after quercetin treatment and reported an oscillation of TER. The study was carefully designed and described in detail. The results that the authors demonstrated here are  scientifically sound and valuable for further medical or nutraceutical applications as the basic character of the famous flavonoid quercetin. Thus, this reviewer basically recommends publishing the manuscript in International Journal of Molecular Sciences. However, unfortunately, this version of the manuscript lacks to refer and discuss some important previous works. Thus, the reviewer recommends adding one brief discussion sections corresponding to the previous works.

Major issues

(1) The reviewer found that the impressively similar quercetin-induced TER oscillation in CACO-2 cell was reported in the following two reports,
“[1] Quercetin enhances intestinal barrier function through the assembly of zonula occludens-2, occludin, and claudin-1 and the expression of claudin-4 in Caco-2 cells. Suzuki T, Hara H. J Nutr. 2009; 139:965-74. doi: 10.3945/jn.108.100867.”, and
“[2] Kaempferol enhances intestinal barrier function through the cytoskeletal association and expression of tight junction proteins in Caco-2 cells. Suzuki T, Tanabe S, Hara H. J Nutr. 2011; 141:87-94. doi: 10.3945/jn.110.125633.”.

Especially, in the figure 3 of the above reference [1], mostly same profile of time-dependent TER change, in which rapid increase at 6 hr, fast decrease at 20 to 24 hr, and moderate re-increase at 48 hr were demonstrated, even in Caco-2 cell, that is distinct to this report. Since the cell type is too different, the reviewer does not think that this manuscript lacks novelty. In addition, this reviewer can believe that this manuscript may contain very general pharmacological action of quercetin not only to kidney cells but also the other epithelial cells. This reviewer suggests that the author may better discuss these points and can improve the generality of their interesting finding.

(2)  Addition to the previous two reports, Piegholdt also reported the temporal 120-140% increase of TER in CACO-2 cell system by quercetin. It can be also referred.
“ Biochanin A and prunetin improve epithelial barrier function in intestinal CaCo-2 cells via downregulation of ERK, NF-κB, and tyrosine phosphorylation. Piegholdt S et al, Free Radic Biol Med. 2014 70: 255-64. doi: 10.1016/j.freeradbiomed.2014.02.025.”.

(3)  According to the mechanism of limited TER increase after 48 hour or longer exposure to quercetin in MDCK II cells. In Figure 3, the authors showed more than 80% loss of claudin 2 after quercetin treatment. Tokuda and Furuse reported that the genetic knockout of claudin-2 in MDCK cells showed significant increase of TER to 2000-3000 Ω x cm2 (PLoS ONE 10(3): e0119869. doi:10.1371/journal. pone.0119869). It was explained that the pore-forming claudin 2 was replaced by the other barrier forming claudins. In this manuscript, although some claudins increased to compensate the loss of claudin-1 and claudin-2, the increase of TER is so limited. How the authors explain?

(4) According to the issue (3), since the increase of TER was so limited, the authors may better show that the integrity of tight junction of MDCK cells is still being preserved after the long-time exposure of quercetin. For example, addition of the data of lucifer-yellow efflux assay is recommended.  This is only the recommendation and not necessary if the authors have an appropriate reason.

Round 2

Reviewer 1 Report

Dear authors,

This is an impressive study with a lot of data presented. Thank you for addressing most of my concerns and for editing the manuscript and being transparent with your data! I consider most of your changes sufficient. Below are my responses to several rebuttals I would like for you to consider further:

Authors’ rebuttal: We also performed a two-way ANOVA grouping all of our actin blots together. Although there is variability, there was no significant difference with time or treatment. We have included this data below.

My response: I’m not sure if this is the best way to demonstrate no variation in actin. Do you mean you averaged: (i) all the actin blots in the paper, or (ii) all the blots in every experiment separately? The latter would be an appropriate way to demonstrate no change. But if you average all actin blots in the paper, it is irrelevant as you’re not using averaged actin values from all plots to normalize your data to.

Authors’ rebuttal: For the immunofluorescence analyses, ZO1 is not used as a control to determine relative levels of claudins. Rather it is used to assess localization of claudins to the tight junction barrier. For measuring co-localization we used the M2 value and we did not correct for intensity. M2 is a measure of how many of the red pixels (ZO1) coincide with green pixels (claudin). M2 is not a reading of abundance or intensity levels and therefore it only informs about localization and it is not a quantification tool. We have clarified this in the manuscript. Again, we used Two-way ANOVA to correctly interpret our data. Furthermore, p-values corresponding to time, treatment and interaction have been reported. With respect to expression levels, we are only able to make qualitative statements based on the immunofluorescence data. We hope that we have made this clear in the manuscript.

My response: I appreciate the explanation and the changes. My initial concern stemmed from the fact that the M2 value would change if the number of red pixels would change (corresponding with changes in ZO1, not the changes in cldns). However, since the quantitative statements have been removed, no further discussion needs to take place.

Authors’ rebuttal: In the corrected manuscript, the graphs have been changed by removing the fold analysis and including the control data. For the new analyses, in order to be able to compare between different blots, each band has been normalized to control at claudin 1h and actin. Also, we have shown the individual data points rather than bar graphs to illustrate individual and independent experiments, and where possible, we have increased the sample size.

My response: I appreciate this change and commend the authors on being transparent with their data!

My initial comment: Section 4.3 - Was background resistance of an empty insert subtracted from your TER measurements?

This is not necessary with the CellZscope machine, if the cells are not 100% confluent in the insert, the electricity can pass without any resistance and the TER measurement is equal to zero. Therefore, the background from an empty insert is zero as well.

My response: This makes no sense to me – according to my experience, electricity does not pass through anything without any resistance – every electrical path has to have resistance unless you are working with super-conductors. In my experience, Corning inserts have background resistance of ~110 Ohm*cm2 when no cells are present on them at all (I have measured this consistently over the last 10 years). Is there a procedure in your protocol that accounts for this – is the machine zeroed with empty inserts prior to seeding cells into the inserts? Any other way to account for the resistance of an empty insert? For a high-resistance culture, this may not be an issue, but your TER values are pretty low, so accounting for background resistance will increase resolution of presented results.  

Also, what model of CellZScope system was used in this study? There are several models available currently.

Other than these minor corrections, data analysis in the ms are much improved!
